# Sexual Desire and Erotic Fantasies Questionnaire: Validation of the Erotic Fantasy Inventory Scale (SDEF3) in Italian Adults

**DOI:** 10.3390/healthcare11060880

**Published:** 2023-03-17

**Authors:** Filippo Maria Nimbi, Roberta Galizia, Lilybeth Fontanesi, Seray Soyman, Emmanuele Angelo Jannini, Chiara Simonelli, Renata Tambelli

**Affiliations:** 1Department of Dynamic, Clinical and Health Psychology, Sapienza University of Rome, 00185 Rome, Italy; 2Department of Psychological, Health and Territorial Sciences, G. D’Annunzio University of Chieti-Pescara, 66100 Chieti, Italy; 3Department of Systems Medicine, University of Rome Tor Vergata, 00133 Rome, Italy

**Keywords:** sexual desire, erotic fantasy, sexual fantasy, questionnaire, psychometric properties

## Abstract

Background: Erotic fantasies are the most common sexual experiences and provide valuable clinical material for understanding individual and relational emotional dynamics. The primary objective of this study is to validate the Sexual Desire and Erotic Fantasies questionnaire (SDEF) Part 3–Inventory of Erotic Fantasies. This questionnaire was designed to be a sex-positive and inclusive measure of the content of erotic fantasies, accessible to individuals of all gender identities, sexual orientations, relationship/romantic status, and sexual behaviors. Methods: The SDEF3 was completed by 1773 Italian participants (1105 women, 645 men, and 23 participants identifying as other genders). Two factorial structures were presented and discussed: a 20-dimension structure for clinical and explorative use and a 6-dimension structure for research purposes. Results: The six-factor version was preferred due to its robust statistical properties and its ability to differentiate between sexually clinical and functional men and women, based on cut-off scores from the Female Sexual Function Index (FSFI) and the International Index of Erectile Function (IIEF). Differences in the frequency of themes in fantasies between gender and sexual orientation were reported and discussed. Conclusions: The current study indicates that the SDEF3 is a valuable and comprehensive measure for assessing various scenarios related to fantasizing activity. It has potential applications in both clinical practice and scientific research.

## 1. Introduction

Sexual fantasies (SF) are usually defined as subjective mental imagery and thoughts that are erotic or arousing to the individual while awake [1]. SF are usually used to arouse the individual and are distinct from daydreaming, which is a spontaneous and fanciful series of thoughts not connected to immediate reality and the need for satisfaction [2]. Erotic fantasizing is regarded as the most common sexual experience among humans [3,4]. Across studies, about 90–97% of the general population report having SF and use them to stimulate their desire and intensify their arousal [1,5,6,7,8,9,10]. Fantasizing is typically referred to as a positive experience that can incentivize sexual response, pleasure, and satisfaction [4,11,12,13,14]. Sharing SF within a relationship may increase the positive perception of the relationship and foster intimacy [14]. However, SF may also represent a negative experience when perceived as unwanted and distressful for the individual [7,15]. SF should not be interpreted as a direct sign of real interest in the behaviors [16,17]. In this sense, some studies have focused on paraphilic interests and kinky SF to evaluate psychosocial factors associated with paraphilias, sexual violence, and deviance. The literature agrees that having SF related to paraphilic themes in the general population is neither rare nor directly connected to committing a crime [18,19]. The presence of paraphilic interests does not seem to be problematic per se (in line with DSM and ICD criteria); having a variety of different SF, rather than specific contents, seems to be a good marker of higher desire/arousal rates and better sexual health outcomes [7,17]. Historically, the literature has emphasized gender differences, with men reporting more SF about partner diversity, visually explicit descriptions, and dominating behaviors, whereas women report more about romance and submission [1,3].

A more complex and nuanced picture of gender differences has emerged from recent studies [7,8,9,10,17,20,21,22], accounting for individual variability and different sexual orientations and identities. In this sense, many studies [8,23,24,25,26,27,28,29] have attempted to categorize SF and create clusters focusing on erotic scenarios that refer to similar patterns (e.g., situations, sexual practices, and objects), following the evolution of the most common SF contents in different societies and cultures.

Focusing on validated measures of erotic fantasies, a masterful contribution dates back to the “Daydreaming inventory for married women” by Hariton and Singer [23]. In their study, the authors explored the incidence of 15 SF in a group of partnered women; however, despite the great contribution to the research, a real adaptation of the tool for the general population was never reported, making the tool obsolete today even for married women.

The Sex Fantasy Questionnaire (SFQ) [25,26] is among the most used questionnaires assessing the content of SF. With 40 items, it describes four dimensions: Exploratory (e.g., group sex, mate-swapping), Intimate (e.g., kissing, oral sex, masturbation), Impersonal (e.g., sex with strangers, voyeurism, fetish), and Sadomasochistic (e.g., whipping and spanking, being forced to have sex). Although it has been widely used, the SFQ misses some relevant updates to reflect contemporary erotic imageries. In fact, a recent study [7] has suggested a modified version of the SFQ. No validation study of this SFQ modified version is available in the literature, although a suggested six-factor structure was proposed by Dyer et al. [30] and used by Allen et al. [31].

O’Donohue et al. [27] created the Paraphilic Sexual Fantasy Questionnaire (PSFQ) to investigate sexual fantasies corresponding to paraphilic interests in accordance with DSM IV criteria, and tested it on a group of convicted male child molesters. Starting from Wilson’s work [25], the PSFQ includes 162 items on topics such as consenting adult partners, masturbation, bondage, sadism, masochism, rape, pedophilia, and other paraphilias. However, both Wilson’s SFQ [25] and O’Donohue et al.’s PSFQ [27] measures have been criticized as too vague, deficient in reliability, and too focused on paraphilias [32].

More recently, Bogaert et al. [28] presented a new measure named (again) Sexual Fantasy Questionnaire (SFQ) composed of 62 items including relevant themes extracted from the literature (e.g., irresistibility, commitment vs. non-commitment, explicitness, roughness/coercion, romantic gestures/situations, and dominance/submission). The participants were asked to indicate their arousal to the items on a 7-point scale (1 = not at all exciting to 7 = extremely exciting). All the items are formulated in heterosexual wording, and participants are asked to substitute the most appropriate pronouns in case they do not find themselves in the formulation. This makes the questionnaire easy to administer, yet at the same time requires an adaptation effort on the part of the participant, which could be tiring and disrespectful.

Two of the most updated measures of SF and related behaviors were developed by Brown and colleagues [33] using a modern statistical approach in a large nonclinical convenience sample of people from the U.S., Canada, U.K., and Ireland. Both measures showed paraphilic and normophilic interests, with higher scores in men and non-heterosexual participants. However, among the limitations of these measures, interests related to pedophilia, fetishism, and other less common paraphilias were not retained in the final version of the measure.

On the Italian population, Panzeri et al. [29,34] validated the Erotic Imagery Questionnaire (EIQ), an 81-item questionnaire exploring five factors: transgressive themes, emotional romantic topics, dominance/submission, variety of partners, and explicit sexual images. Despite the acknowledgement of the thematic updates, Nese et al. [22] suggested that the EIQ dimensions collect sexual fantasies based on general characteristics, with the limitation that very diverse fantasies may collide in the same statistical dimension. This makes the operationalization of the dimensions on a clinical and research level very difficult. The labels used to explain the themes are not accurate enough to offer a good description of the sexual fantasy content. Therefore, Nese et al. [22] call for the need for a measure that is easier to use and interpret.

On a general level, the content of sexual fantasies is difficult to measure and categorize, as it relies on self-reports that are influenced by social desirability and other variables [32]. These measurement challenges, also considering data representability and questionnaire reliability, have contributed to the debate on how best to assess sexual fantasies in sex research [35]. Keeping in mind that the self-report limitation cannot currently be overcome in this field, except by associating measures of social desirability during the assessment, available tools present some other relevant problems. Sexual fantasies are extremely important and useful clinical material, irrespective of the approach, that allows experiencing and being aware of individual and relational emotional dynamics [12]. In this sense, an inventory that is up-to-date and comprehensive can be a useful support to the clinician, both as a stimulus that can be used within individual or couple therapies, and as a tool to elicit reflections, comparisons, and open communication on intimate topics. The aforementioned questionnaires, as well as other questionnaires in the literature, represent diversified attempts to categorize complex and ever-changing erotic scenarios across time and cultures. Firstly, questionnaires on the contents of SF do not always pursue the same goal [22]: some of them focus on the frequency of SF, others evaluate the arousing potential of the presented scenarios, while others focus on how typical/pathological the contents might be. Another limitation, according to Cartagena-Ramos et al. [36], is that questionnaires on SF are not often applied to different sociocultural contexts than that of their authors, with significant issues regarding the replicability of the results and reliability. Moreover, many measures are limited to the heterosexual cisgender population, failing to capture other possible sexual identities (gender, sexual orientation, and other expressions) and behaviors [9,10].

### 1.1. The Current Study

The current study is inserted in this complex scenario as a part of a wider project aiming to provide an updated, extensive, and inclusive measure of SF that can be easily adapted to different cultural backgrounds and used in clinical work in a second phase of the project (cross-cultural validation). The project aims to analyze the psychometric properties of a composite measure for sexual desire called the “Sexual Desire and Erotic Fantasies questionnaire (SDEF)”. The SDEF is divided into three independent measures (1. Sexual Desire; 2. Use of Erotic Fantasies; and 3. Erotic Fantasies Inventory) [37,38] that can be used separately or together for a general overview of the desire function. The creation of the SDEF was driven by the need to have a tool able to explore different aspects of the desire experience rather than to improve the currently available measures. Especially, it was designed to be used in clinical settings for the investigation of key components that should be observed in the assessment of sexual dysfunctions and relational problems as highlighted by the major diagnostic classifications such as DSM-5 and ICD-11 [39,40].

In this paper, we will test and discuss the results of the validation study of the Sexual Desire and Erotic Fantasies questionnaire–Part 3 Erotic Fantasies Inventory (SDEF3), which focuses on the frequency of SF contents. The SDEF3 was created to explore a person’s erotic imaginative experience and to collect a wide range of erotic stimuli based on scientific literature. This variety is considered one of the strengths of the SDEF3, which may help clinicians to assess desire and/or sexual-related difficulties, and can also be used in research to deepen specific characteristics of SF. Furthermore, a sex-positive approach [41] was used to build the SDEF3 as a tool accessible to all individuals, regardless of their gender identities, sexual orientations, relational/romantic status, and sexual behaviors. Specifically, special attention was paid to writing items with inclusive language that is capable of describing different manifestations of human sexuality, such as non-penetrative sexual behaviors. This is particularly important in the Italian language, in which gender binary declinations can create difficulties and misinterpretations. In this sense, a sex-positive approach recognizes the tremendous cultural diversity in sexual practices, acknowledging substantial variations in personal meanings and preferences over time and space.

### 1.2. Aims

The main aim of the present study is to explore the factorial structure of the SDEF3 questionnaire and to discuss its psychometric properties. The first objective focuses on testing the internal reliability, construct, and discriminant validity of a 20-factor structure of the questionnaire. This large and descriptive model aims to illustrate clusters of SF that may vary together. The second objective is to test an alternative 6-factor structure of the SDEF3, which is more concise and has more rigorous statistical criteria, and a reduced number of domains. The internal reliability, construct and discriminant validity of the questionnaire will be evaluated. The third objective is to explore the characteristics of the SDEF3 six-factor structure, such as its association with sociodemographic variables, sexual functioning, and differences in gender and sexual orientation, in a group of Italian individuals.

## 2. Materials and Methods

A total of 1819 volunteers from the Italian general population participated in the SDEF validation study, of which 1135 were women, 661 were men, and 23 identified as other genders. Participants were recruited using a snowball technique and sharing advertisements on institutional websites and social networks such as Facebook, Instagram, and LinkedIn. The web survey was available on the Google.forms platform from January 2019 to December 2020. Participants provided informed consent before accessing the survey, and the questionnaire was anonymous with no remuneration provided. The institutional ethics committee of the Dept. of Dynamic, Clinical and Health Psychology, Sapienza University of Rome, Italy (protocol code 14) approved the study on 9th January 2019. The inclusion criteria were being at least 18 years old and holding Italian citizenship. The present study excluded 46 responses (2.53%) due to duplicated, falsified, or incomplete records. The final group comprised 1773 participants, including 1105 women, 645 men, and 23 other genders. To conduct explorative and confirmative factorial analysis, participants were randomly assigned to two different groups, balanced for gender, age, and sexual orientation (see Table 1). The same group of participants was also involved in the validation study of SDEF1 and SDEF2 [37,38].

In the current study, different measures were involved. Firstly, participants completed a brief sociodemographic questionnaire to provide information about age, gender, sexual orientation, marital and relationship status, children, education level, work status, religious and political orientation, and whether they were sexually active or not.

The Sexual Desire and Erotic Fantasies questionnaire (SDEF3–Part 3) is designed to measure the frequency of the most common SF. The items included in the SDEF3 were created by reviewing and selecting fantasy contents from relevant literature on fantasies and pornography trends [7,8,23,24,25,26,27,28,29,42,43] to offer an inclusive and updated list of erotic situations, practices, and objects. The authors developed 153 items during this process, paying particular attention to the use of inclusive language that could refer to any erotic activity, not only penetrative sex (e.g., kissing, body stimulation, oral sex, masturbation), and trying to be respectful of any gender identity and sexual orientation. Answers are rated using a 5-point Likert scale (from “never” to “always”) to indicate the frequency of SF related to the presented stimuli in the last six months (time frame chosen in line with DSM-5 criteria for desire-related disorders). Higher scores indicate a higher frequency of fantasies. At the end of the questionnaire, a part is left free for the person to indicate fantasies that have not been included in the previous items as open-ended questions. A pool of ten experts in the field of psychosexology and sexual medicine reviewed the content by inserting comments and suggestions into the text separately. The criteria used by the experts were content relevance and comprehension. Once all the comments from the experts had been collected, the authors revised each item by incorporating minor wording changes and merging or deleting 28 items that were redundant (e.g., ‘drinking urine’ or ‘receiving urine on the body’ has been merged into item 78 ‘urine’). The remaining 125-item version of the SDEF3 was pilot-tested with 20 volunteers to examine the general comprehension of the questionnaire and was then administered in the present study to test its psychometric characteristics. The final version is reported in Appendix A.

Sexual Desire Inventory–2 (SDI-2) [44]: The SDI-2 is a 14-item measure used to evaluate two dimensions of sexual desire: dyadic and solitary sexual desire. Higher scores indicate a higher level of sexual desire. The two-dimensional structure presents satisfying psychometric properties, as in the Italian version [45], with a Cronbach’s alpha coefficient in this study equal to 0.88 for dyadic and 0.91 for solitary sexual desire.

International Index of Erectile Function (IIEF) [46]: The IIEF is a widely used 15-item questionnaire for the evaluation of male erectile and sexual function. A general index of sexual function and five specific dimensions are calculated: sexual desire, erectile function, orgasmic function, satisfaction with intercourse, and overall satisfaction. Higher scores indicate better functioning. Psychometric studies have reported good reliability, validity, and discrimination between sexually dysfunctional and healthy people (clinical cut-off score <26). For this study, the IIEF was worded in a such way as to be completed by all cisgender men, regardless of their sexual orientation. The Cronbach’s alpha in this study ranged from 0.87 (orgasmic function) to 0.93 (overall satisfaction).

Female Sexual Function Index (FSFI) [47]: The FSFI is an established 19-item instrument providing information on general sexual functioning and six specific dimensions: sexual desire, sexual arousal, lubrication, orgasm, sexual pain, and sexual satisfaction. Higher scores indicate better functioning. The measure presents good test–retest reliability, internal consistency, validity, and discrimination between sexually dysfunctional and healthy people (clinical cut-off score <26.55), as in the Italian version [48]. For this study, the FSFI was worded in a such a way as to be completed by all cisgender women, regardless of their sexual orientation. The Cronbach’s alpha in this study ranged from 0.81 (sexual arousal) to 0.92 (sexual pain). The calculation of total FSFI/IIEF scores and relevant domain scores (on all but the desire domain) were limited to those who did not indicate a zero score (no sexual activity) on any of the FSFI/IIEF items.

Marlowe–Crowne Social Desirability Scale–Short Form (MCSDS-SF) [49]: The MCSDS-SF is a 13-item measure developed to assess socially desirable responses. Higher scores indicate a higher tendency to respond in a socially desirable way. The Cronbach’s alpha value for this measure was 0.91. The MCSDS-SF was used as a covariate in the analysis of the current study to limit the effects of social desirability.

Regarding statistical analyses, the psychometric properties of the SDEF3 were tested using different procedures. Construct validity was estimated at the item level using principal component analysis (PCA) to identify the underlying constructs of the questionnaires. In this phase, a direct oblimin rotation was used, and the number of factors selected was determined using parallel analysis in conjunction with the Guttman–Kaiser criterion, using Monte Carlo PCA for parallel analysis by Watkins [50]. After establishing a satisfactory model, a path diagram was drawn and tested using confirmatory factor analysis (CFA). Internal consistency was assessed using Cronbach’s alpha. Composite reliability (CR) and average variance extracted (AVE) values were examined. Pearson correlations (two-tailed), one-way and two-way multivariate analyses of covariance (MANCOVAs) were used to explore associations with erotic fantasy dimensions and sociodemographic variables, sexual functioning, gender, and sexual orientation differences in a group of Italian people. Age, relationship status, and social desirability effects were controlled using these as covariates in the MANCOVAs. PCA, Cronbach’s alpha values, Pearson correlations, and MANCOVAs were performed using IBM SPSS 27.0, and CFA was tested with IBM SPSS Amos 22 (Version22, IBM Corp, Armonk, NY, USA).

## 3. Results

Table 1 shows the sociodemographic data for the variables assessed in the study organized for the total group (*n* = 1773) and for the two subgroups randomly extracted to run exploratory and confirmatory factorial analyses separately (Group 1 *n* = 887; and Group 2 *n* = 886).

### 3.1. Testing the SDEF 20-Factor Structure

Group 1 was used to test the factorial structure of the SDEF3 with principal component analyses (PCAs). After excluding open-ended qualitative items (121, 122, 123, 124 and 125), PCAs were run on the remaining 120 items of the SDEF3 using a direct oblimin rotation. A Kaiser–Meyer–Olkin value of 0.92 supported the adequacy of the sample. The significance of the Bartlett test of sphericity (χ^2^ = 100,659.879; *p* < 0.001) meant that item correlations were large enough to conduct PCAs.

Based on eigenvalues higher than 1, 20 components were identified, accounting for 64.34% of the total variance. Item selection was based on loadings higher than 0.3 on respective factors. A total of 23 items loaded below 0.3 in all factors or loaded higher than 0.3 in more than one factor. Thus, they were excluded from the following analyses. Appendix B presents the retained 97 items’ component loadings. Factors highlighted are:F1. Physical Characteristics: Dimension collecting physical characteristics culturally attributed to being beautiful/handsome, attractive, and sexy, such as athletic/thin body, young age, etc.F2. Group sex: Dimension collecting different sexual scenarios having more than one sexual partner involved.F3. Romantic: Dimension gathering romantic scenarios and actions such as kissing, hugging, massage, and looking after/being looked after by a partner.F4. Vanilla Sex: Dimension collecting a range of common sexual practices such as petting, oral sex, masturbation, and vaginal sex.F5. Masochism: Dimension describing a range of torture, humiliating and painful practices received by the participant from other partners.F6. Sadism: Dimension gathering a range of torture, humiliating and painful practices performed by the participant on other partners.F7. Taboo: Dimension gathering taboo scenarios such as having sex with animals, children, corpses, and people with disabilities.F8. Anal Sex and Toys: Dimension collecting a range of activities involving anal play and sex toys.F9. Incestuous/Older people: Dimension collecting fantasies around family members, pregnant women, elderly people, and obese people.F10. Soft Fetish: Dimension describing a range of fetishes such as foot, hair, saliva, sweat and other parts of the body.F11. Risk of Being Caught: Dimension regarding open air scenarios or places in which is easy to be caught by others while having sex.F12. Past Experience: Dimension describing fantasies involving memories of past sexual experiences and former partners.F13. Seduction and Infidelity: Dimension regarding themes in which seduction and betrayal of a relationship are central in the erotic scene.F14. Exhibitionism and Voyeurism: Dimension that describes the activity of watching or being spied on while naked or engaging in sexual activity.F15. Bondage: Dimension describing a range of practices involving the action of tying/being tied up and blindfolded.F16. Sexual Abuse: Dimension regarding scenarios involving non-consensual sexual activities.F17. Sex work: Dimension gathering scenarios where sex is bought or sold, including playing in a porn movie.F18. Ejaculation Emission: Dimension collecting groups of fantasies in which the person ejaculates on the partner.F19. Receiving Ejaculation: Dimension collecting groups of fantasies in which the person receives the ejaculation of the partner.F20. Dirty Fetish: Dimension describing liquids fetishes such as urine, excrement, and vomit.

To validate the 20-factor structure, a CFA was run on Group 2 to measure model fit, comparison, and parsimony indices, following the procedure used by Nimbi et al. [51,52]. The figure of the model was not reported for simplicity. The maximum likelihood estimation method was used, and pathways of error variance between items inside the same factor were inserted to increase model fit. The χ^2^ value for the model was significant (χ^2^ = 18,012.66, *p* < 0.001), and the RMSEA was 0.043 (90% CI = 0.042–0.043). Other fit indices evaluated included GFI (0.79), NFI (0.82), and CFI (0.86). Moderate fit was reached in all measures except for the χ^2^ value, which is sensitive to large sample sizes (*n* > 200).

Regression coefficients for this model ranged from 0.27 to 0.94 and were all statistically significant (*p* < 0.001). MacDonald’s omega coefficients for internal consistency were satisfactory, ranging from 0.65 (F9. Incestuous/Older people) to 0.91 (F18. Ejaculation Emission). The composite reliability (CR) for each construct was above or close to the expected threshold of 0.70 (F1 = 0.83; F2 = 0.86; F3 = 0.85; F4 = 0.84; F5 = 0.82; F6 = 0.59; F7 = 0.82; F8 = 0.76; F9 = 0.65; F10 = 0.67; F11 = 0.85; F12 = 0.73; F13 = 0.73; F14 = 0.76; F15 = 0.85; F16 = 0.67; F17 = 0.34; F18 = 0.81; F19 = 0.82; F20 = 0.67). F6 and F17 showed a low CR. The average variance extracted (AVE) value for each factor was below the expected threshold of 0.50 for most of the factors (F1 = 0.33; F2 = 0.48; F3 = 0.45; F4 = 0.45; F5 = 0.44; F6 = 0.23; F7 = 0.47; F8 = 0.39; F9 = 0.28; F10 = 0.3; F11 = 0.6; F12 = 0.41; F13 = 0.41; F14 = 0.45; F15 = 0.58; F16 = 0.4; F17 = 0.14; F18 = 0.59; F19 = 0.61; F20 = 0.41), except for F11, F15, F18, and F19.

Intercorrelations between the 20 factors for the total group (*n* = 1773) are reported in Table 2. The 20-factor structure presented some strengths, such as the ability to describe different scenarios of erotic fantasies, and important weaknesses, such as fair psychometric characteristics, which will be discussed later. Therefore, the 20-factor structure was discarded, and it was decided to test a structure with six factors.

### 3.2. Testing the SDEF3 Six-Factor Structure

In line with the objectives of this study, a more robust factorial structure using more rigorous criteria for factor extraction was tested. A new set of PCAs was run on the 120 quantitative items of the SDEF3 using a direct oblimin rotation. Monte Carlo parallel analysis identified six components accounting for 47.9% of the total variance. Item selection was based on loadings higher than 0.4 on respective factors. A total of 54 items loaded below 0.4 in all factors or loaded higher than 0.4 in more than one factor. Thus, they were excluded from the following analyses. Appendix C presents the retained 66 items’ component loadings. The factors highlighted were:F1. Physical and Contextual: A dimension that collects a series of physical characteristics inspired by common culturally widespread canons of beauty (e.g., athletic/thin body, tall, young age) and places or scenarios considered erotically stimulating and representative in mainstream pornography (e.g., outdoor sex, seduction, having sex at work).F2. BDSM: A dimension collecting different sexual scenarios that recall BDSM and bondage practices, sadomasochistic activities, fetishism, and similar.F3. Taboo: A dimension gathering taboo scenarios such as having sex with animals, children, relatives, corpses, and rape among others.F4. Bottom: A dimension collecting a range of common sexual practices in which the person plays the role of receiving the practice (bottom/passive) with a partner who plays a more leading/active role.F5. Top: A dimension collecting a range of common sexual practices in which the person plays the role of doing the practice (leader/active) with a partner who plays a more passive/bottom role of receiving it.F6. Romantic: A dimension gathering romantic scenarios and actions such as kissing, hugging, massage, and looking after/being looked after by a partner.

To validate the six-factor structure identified with the PCA, a CFA was run on Group 2 measuring model fit, comparison, and parsimony indices. For simplicity, the figure of the model is not reported. The maximum likelihood estimation method was used. To increase model fit, pathways of error variance between items inside the same factor were inserted. The χ^2^ value for the model was significant (χ^2^ = 12,067.76, *p* < 0.001). RMSEA was 0.046 (90% CI = 0.045–0.047). Other fit indices evaluated included GFI (0.87), NFI (0.89), and CFI (0.91). A better fit was reached in all measures except for the χ^2^ value due to its sensitivity to large sample sizes (*n* > 200) compared to the 20-factor structure.

Regression coefficients for this model ranged from 0.37 to 0.88 and were all statistically significant (*p* < 0.001). Internal consistency was assessed: MacDonald’s omega coefficients were satisfactory (F1 = 0.88; F2 = 0.9; F3 = 0.8; F4 = 0.85; F5 = 0.87; F6 = 0.83); the CR for each construct was above the expected threshold of 0.70 (F1 = 0.88; F2 = 0.88; F3 = 0.85; F4 = 0.85; F5 = 0.86; F6 = 0.87); the AVE value for each factor was below the expected threshold of 0.50 (F1 = 0.28; F2 = 0.37; F3 = 0.32; F4 = 0.43; F5 = 0.47; F6 = 0.49).

### 3.3. SDEF3 Six-Factor: Testing Validity Evidence Based on the Relationship with Other Variables

Based on the total group (*n* = 1773), intercorrelations between the six factors were all statistically significant (Table 3). Table 3 also reports Pearson’s correlations with SDI-2, FSFI, and IIEF scores to verify associations with desire and other sexual domains. Focusing on an SDEF3 six-factor description, associations with sociodemographic variables were explored. Table 4 reports Pearson’s correlations with age, being in a relationship, education level, political and religious attitudes, having sexual intercourse (being sexually active), and social desirability. Different erotic contents were shown to be significantly associated with sociodemographic variables such as age, relationship status, having children, education level, having sexual intercourse, and political and religious attitudes.

Due to the importance highlighted in the current results and similar constructs in literature, age, relationship status, and social desirability were considered as covariates in the following analyses aiming to explore possible differences in the contents of erotic fantasies among genders and sexual orientations [9,10]. Due to the limited number of transgender/gender-nonconforming, asexual, and pansexual participants, the following analyses focused on people identifying themselves as women and men (gender) and heterosexual, bisexual, or homosexual (sexual orientation).

A two-way MANCOVA (having age, being in a relationship, and social desirability as covariates) was run to highlight gender and sexual orientation differences on SDEF3 factors. Gender and sexual orientation were considered as independent variables, while SDEF3 dimensions were put as dependent ones. Findings are reported in Table 5, showing significant results for gender, sexual orientation, and gender X sexual orientation (Figure 1).

To explore whether the SDEF3 six dimensions were able to differentiate between clinical scores of FSFI and IIEF, two one-way MANCOVAs (having age, being in a relationship, and social desirability as covariates) were run to highlight sexual functioning differences on SDEF3 factors. Reaching a clinical score of FSFI for women and IIEF for men was considered as an independent variable, while SDEF3 dimensions were put as dependent ones. Findings are reported in Table 6, showing significantly higher SF in all SDEF3 dimensions for participants with FSFI and IIEF functional scores compared with those with clinical scores, except for F3 in both genders. These results seem to suggest the ability of the SDEF3 to discriminate among sexually functional and dysfunctional men and women.

## 4. Discussion

The current study aimed to test the psychometric properties of a self-administered measure of sexual fantasies (SF). Two structures were tested: a 20-factor structure that was able to describe a wide variety of erotic themes with fair psychometric characteristics, and a 6-factor structure with more reliable psychometrics.

The 20-factor version included 97 items explaining 64.34% of the total variance. The strengths of this version concern the possibility of describing a great variety of themes and scenarios that could be useful in clinical and explorative/descriptive contexts. Therapists may use the list of SF (dimensions and items) to help their patients reflect on their erotic repertoire, identify which elements they like or dislike, and communicate with their partners [12,52]. Having more awareness of one’s fantasies might be useful for dealing with desire issues and other problems in the sexual and relational sphere [14]. This allows, especially in the clinical setting, for discussing together how the individual or the partners communicate sexually about wishes, boundaries, and how sexuality is negotiated [52]. Furthermore, it may be easier to collect relevant information about SF by avoiding direct disclosure during the clinical interview. However, it should be noted that any form of categorization of fantasies, as with other natural phenomena, results in a simplification of reality and a loss of information [53]. Therefore, the SDEF3 should be considered a starting point to stimulate creativity and free-thinking, rather than a rigid model to follow. As critical elements of the 20-dimension version, we underline the low statistical power of some psychometric indexes, especially the AVE. Furthermore, using a tool that includes 20-dimensions could be difficult to operationalize in research for qualitative studies. For this reason, the 20-factor structure is not reliable enough to be recommended for scientific use. In the process of this study, it was essential to test a different (six-factor) structure based on more rigorous and stable statistical criteria.

The six-factor structure includes 66 items that explain 47.9% of the total variance. It provides general categories that may be comparable to those used in other studies [8,25,26,28,29]. Although having a categorization based on six dimensions can be reductive compared to the 20 previously presented, it may allow for an easier operationalization of SF. In this sense, the relevance of these broad categories was explored in relation to sociodemographic and sexual functioning variables.

Regarding associations with sociodemographic data, physical and contextual, BDSM, and romantic fantasies showed lower frequency with aging, being in a relationship, having children, and higher education levels. A major political involvement (regardless of the type of party) seemed to relate to a higher frequency of physical and contextual, BDSM, and top fantasies. Religiousness seemed to be connected to reporting fewer SF of any kind. Social desirability showed a relationship with all SDEF3 dimensions except for romantic fantasies, which might be intended as more “morally” acceptable as an expression of love and intimacy [54]. These results are in line with previous studies [55,56,57,58].

Sexual intercourse frequency showed a significant relationship with BDSM, bottom, and top fantasies. As expected [14], all the erotic dimensions were linked to a higher level of desire reported, both at the dyadic level and regarding masturbation, with the only exception of romantic fantasies, which do not seem to characterize solitary sexual activity. Regarding other phases of sexual response and satisfaction, in women, BDSM, bottom, and top fantasies seem to relate to higher FSFI scores, while in men, only top fantasies show a positive correlation with the IIEF scores. These results are not surprising and are in line with leading western gender scripts [28,58].

Moreover, for a clinical application of these results, it should be noted that the presence of specific SF could be a central expression of functional and satisfying sexuality. On the other hand, a problem in sexual functioning may also negatively influence fantasizing activity [53,59]. Therefore, in the case of sexual difficulties, the area of desire and fantasies should be investigated by the clinician for a deeper insight into the problem.

Regarding gender differences in fantasizing, controlling age, relationship status, and social desirability’s effects, men significantly reported a higher frequency of physical and contextual, taboo, and top fantasies than women. No differences were found regarding other SDEF3 domains. In line with other studies [7,22,25,29,60,61], gender differences in SF were expected, although the effect size was small (physical and contextual, taboo fantasies) to medium (top fantasies). Wilson [25] showed that men reported about twice more SF than women, especially in exploratory and impersonal categories, according to the fact that men’s SF seemed to be more explicit (and pornographic) than women’s. Joyal et al. [7] highlighted how men usually reported more different themes than women. In any case, we should consider that in the almost 35 years since Wilson’s studies, many things could have changed in line with the sociocultural evolution on sexuality that favors a greater openness to these issues and influences what we find to be erotic and what we fantasize about [41].

Considering sexual orientation, heterosexual participants seem to report significantly lower scores on physical and contextual, BDSM, and bottom fantasies compared to bisexual and homosexual participants. Moreover, bisexual participants reported a higher frequency of BDSM fantasies than homosexual participants. Intersecting gender and sexual orientation, gay men reported a higher frequency of physical and contextual and bottom fantasies and lower rates of top fantasies compared to other men, while bisexual women reported a higher frequency of physical and contextual, BDSM, bottom, and top fantasies compared to other women. Heterosexual and lesbian women reported lower scores in most of the domains assessed, supporting the idea expressed by Nese et al. [22] that they (1) may report fewer sexual fantasies in general, (2) may express less diversified contents (mainly romantic ones), and (3) the contents of their fantasies might be poorly represented by the items used in this study. In any case, these results are extremely important as they add data to the scarce literature on sexual minorities, especially bisexual men and lesbian women [9,10,62].

Another important result regards the ability of the SDEF3 six-factor solution to differentiate between sexually dysfunctional and functional women and men. Table 6 shows how groups of women and men having clinical scores on the FSFI and IIEF [46,47] got significantly lower scores in all the SDEF3 domains except for taboo fantasies. These results are in line with the idea that SF do not compensate for sexual deficiencies [1,14]. However, future studies should explore whether both the frequency and content of SF may compensate for the overall relational distress rather than for sexual dissatisfaction per se [14]. In any case, the literature agrees that SF are typically involved in the promotion of sexual arousal, pleasure, and satisfaction [1,4,8,11,12,13,14].

As shown by the current results, more frequent fantasizing is associated with higher scores on desire, arousal, orgasm, and satisfaction. To promote relationship intimacy and improve sexual functioning, many clinicians propose SF training to their patients [12,14,63]. Sexual fantasizing may offer individuals/couples a space to work on sexual communication, creating an opportunity to learn to talk about themes, possible experiences, and desires that are typically welcomed by patients [53].

The present research has some limitations that should be highlighted for the reader. (i) Participants were selected using a “snowball” technique; therefore, it is not possible to generalize the results to the Italian population, despite the large number of participants involved in the current study. (ii) The SDEF3 was created as an inventory of themes and fantasy scenarios selected among the most common and frequent ones in the literature. In this sense, they may have guided the participants in their choices, limiting free expression or facilitating falsification. Therefore, any assertion on people’s real fantasizing activity should be made with extreme caution. To limit this bias, the study used a large group of participants, giving them the possibility to report themes or scenarios not covered by the questionnaire with open answers and a social desirability measure that was assessed. However, the group assessed showed limited gender diversity. (iii) Test–retest reliability was not assessed in the current study. For that reason, further studies should be conducted to replicate the present findings and extend the psychometric understanding of the SDEF3. Moreover, future studies should consider extending the evaluation of SF to different sexual identities and orientations beyond binarism. Multicultural studies on the SDEF3 psychometric properties and, more generally, on SF to explore differences and similarities between countries are also needed, highlighting the importance of capturing potential changes over time linked to sociocultural factors [22,61].

## 5. Conclusions

Erotic fantasizing activity remains a complex and largely unknown area of investigation; however, studies such as the present one may help take a small step forward. Specifically, the current study extends the current knowledge about the frequency of specific themes in SF and their connections with sexual functioning among genders and sexual orientations. This may be important not only for advances in research but also for improvements in clinical practice [9,10,11,12]. Sexual therapists should acknowledge the role played by SF and use specific techniques in their clinical practice to improve sexual functioning, sexual communication, relational intimacy, and satisfaction [12,41]. For this purpose, the SDEF3 six-factor solution could be a useful and valid measure to assess different expressions of erotic repertoires for clinical and research purposes. Moreover, we suggest assessing the SDEF3 six-factor solution in association with the SDEF1 and SDEF2 [37,38] in order to have a more complex view of the sexual phantasmatic experience and expression of individuals.

## Figures and Tables

**Figure 1 healthcare-11-00880-f001:**
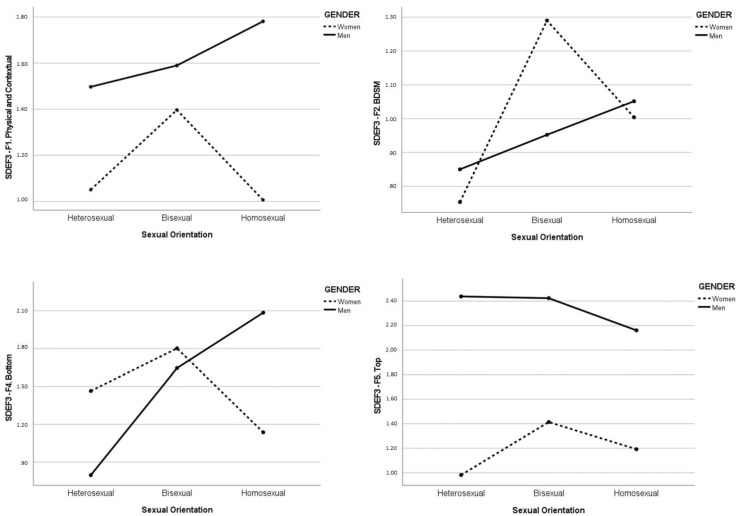
Diagrams of Gender X Sexual Orientation on SDEF3 six-factor solution (MANCOVAs) (*n* = 1729).

**Table 1 healthcare-11-00880-t001:** Sociodemographic variables description.

Variables		Group 1 (*n* = 887)	Group 2 (*n* = 886)	Total Group (*n* = 1773)
		M ± ds (Min–Max)	M ± ds (Min–Max)	M ± ds (Min–Max)
Age		29.3 ± 10.42 (18–78)	29.32 ± 10.28 (18–65)	29.31 ± 10.35 (18–78)
		***n* (%)**	***n* (%)**	***n* (%)**
Gender	Female	555 (62.57)	550 (62.08)	1105 (62.32)
	Male	320 (36.08)	325 (36.68)	645 (36.38)
	Transgender	3 (0.34)	3 (0.34)	6 (0.34)
	Non-binary	9 (1.01)	8 (0.91)	17 (0.96)
Sexual Orientation	Heterosexual	705 (79.48)	703 (79.35)	1408 (79.41)
	Bisexual	80 (9.02)	82 (9.26)	162 (9.14)
	Homosexual	89 (10.03)	89 (10.05)	178 (10.04)
	Asexual	10 (1.13)	9 (1.02)	19 (1.07)
	Pansexual	3 (0.34)	3 (0.34)	6 (0.34)
Marital Status	Unmarried	763 (86.02)	736 (83.07)	1499 (84.55)
	Married	96 (10.82)	125 (14.11)	221 (12.46)
	Separated	24 (2.71)	24 (2.71)	48 (2.71)
	Widowed	4 (0.45)	1 (0.11)	5 (0.28)
Relationship Status	Single	333 (37.54)	293 (33.07)	626 (35.31)
	Couple	532 (59.98)	576 (65.01)	1108 (62.49)
	Polyamory	22 (2.48)	17 (1.92)	39 (2.2)
Children	No	787 (88.73)	764 (86.23)	1551 (87.48)
	Yes	100 (11.27)	122 (13.77)	222 (12.52)
Education Level	Middle School	19 (2.14)	21 (2.37)	40 (2.26)
	High School	286 (32.24)	333 (37.58)	619 (34.91)
	University	443 (49.94)	396 (44.7)	839 (47.32)
	PhD and Postgrad courses	139 (15.67)	136 (15.35)	275 (15.51)
Work Status	Student	422 (47.58)	414 (46.73)	836 (47.15)
	Employed	241 (27.17)	274 (30.93)	515 (29.05)
	Freelance	150 (16.91)	140 (15.8)	290 (16.36)
	Unemployed	64 (7.22)	56 (6.32)	120 (6.77)
	Retired	10 (1.13)	2 (0.23)	12 (0.68)
Sexual Intercourse in Life	Never	45 (5.07)	54 (6.09)	99 (5.58)
	Yes	842 (94.93)	832 (93.91)	1674 (94.42)
Sexual Intercourse in the last 6 months	No	138 (15.56)	110 (12.42)	248 (13.99)
	Yes	749 (84.44)	776 (87.58)	1525 (86.01)

**Table 2 healthcare-11-00880-t002:** Person’s correlation matrix among SDEF3 domains of the 20-factor solution.

	F1	F2	F3	F4	F5	F6	F7	F8	F9	F10	F11	F12	F13	F14	F15	F16	F17	F18	F19	F20
F1. Physical Characteristics	1	.																		
F2. Group sex	0.433 **	1																		
F3. Romantic	0.171 **	0.001	1																	
F4. Vanilla Sex	0.263 **	0.290 **	0.444 **	1																
F5. Masochism	0.224 **	0.448 **	0.075 *	0.247 **	1															
F6. Sadism	0.425 **	0.424 **	0.147 **	0.303 **	0.480 **	1														
F7. Taboo	0.274 **	0.189 **	0.081 **	0.058 †	0.215 **	0.254 **	1													
F8. Anal Sex and Toys	0.317 **	0.462 **	0.134 **	0.434 **	0.389 **	0.370 **	0.129 **	1												
F9. Incestuous/Older people	0.530 **	0.383 **	0.078 *	0.174 **	0.286 **	0.308 **	0.407 **	0.270 **	1											
F10. Soft Fetish	0.500 **	0.329 **	0.202 **	0.266 **	0.351 **	0.447 **	0.193 **	0.356 **	0.355 **	1										
F11. Risk of Being Caught	0.363 **	0.432 **	0.176 **	0.357 **	0.378 **	0.370 **	0.115 **	0.319 **	0.279 **	0.310 **	1									
F12. Past Experience	0.380 **	0.272 **	0.284 **	0.275 **	0.201 **	0.278 **	0.163 **	0.193 **	0.269 **	0.295 **	0.296 **	1								
F13. Seduction and Infidelity	0.443 **	0.373 **	0.220 **	0.301 **	0.208 **	0.280 **	0.128 **	0.236 **	0.338 **	0.266 **	0.389 **	0.449 **	1							
F14. Exhibitionism and Voyeurism	0.388 **	0.580 **	0.075 *	0.250 **	0.348 **	0.344 **	0.211 **	0.409 **	0.339 **	0.349 **	0.463 **	0.217 **	0.320 **	1						
F15. Bondage	0.232 **	0.401 **	0.228 **	0.399 **	0.536 **	0.533 **	0.111 **	0.354 **	0.165 **	0.298 **	0.431 **	0.240 **	0.218 **	0.305 **	1					
F16. Sexual Abuse	0.279 **	0.312 **	0.074 *	0.098 **	0.303 **	0.346 **	0.423 **	0.228 **	0.347 **	0.264 **	0.252 **	0.200 **	0.223 **	0.408 **	0.205 **	1				
F17. Sex work	0.434 **	0.543 **	0.075 *	0.182 **	0.367 **	0.337 **	0.260 **	0.338 **	0.385 **	0.351 **	0.381 **	0.263 **	0.316 **	0.529 **	0.268 **	0.429 **	1			
F18. Ejaculation Emission	0.485 **	0.356 **	0.133 **	0.370 **	0.054 †	0.422 **	0.109 **	0.473 **	0.318 **	0.351 **	0.268 **	0.204 **	0.241 **	0.322 **	0.167 **	0.191 **	0.290 **	1		
F19. Receiving Ejaculation	0.125 **	0.310 **	0.177 **	0.352 **	0.479 **	0.144 **	0.087 **	0.451 **	0.121 **	0.249 **	0.336 **	0.122 **	0.152 **	0.278 **	0.340 **	0.111 **	0.236 **	0.181 **	1	
F20. Dirty Fetish	0.216 **	0.289 **	0.007	0.042	0.341 **	0.297 **	0.254 **	0.288 **	0.330 **	0.414 **	0.164 **	0.085 **	0.085 **	0.298 **	0.140 **	0.213 **	0.275 **	0.182 **	0.198 **	1

Note: † = *p* < 0.05; * = *p* < 0.01; ** *p* < 0.001.

**Table 3 healthcare-11-00880-t003:** Person’s correlation matrix between SDEF3 six-factor solution, SDI-2, FSFI and IIEF (*n* = 1773).

	SDEF3 F1	SDEF3 F2	SDEF3 F3	SDEF3 F4	SDEF3 F5	SDEF3 F6
SDEF3–F1. Physical and Contextual	1					
SDEF3–F2. BDSM	0.487 **	1				
SDEF3–F3. Taboo	0.393 **	0.324 **	1			
SDEF3–F4. Bottom	0.34 **	0.466 **	0.173 **	1		
SDEF3–F5. Top	0.465 **	0.329 **	0.268 **	0.275 **	1	
SDEF3–F6. Romantic	0.227 **	0.199 **	0.094 **	0.206 **	0.213 **	1
SDI-2–Solitary Desire	0.398 **	0.25 **	0.192 **	0.265 **	0.352 **	−0.032
SDI-2–Dyadic Desire	0.49 **	0.291 **	0.189 **	0.324 **	0.461 **	0.265 **
FSFI–Sexual Desire	0.347 **	0.355 **	0.11 **	0.45 **	0.362 **	0.25 **
FSFI–Arousal	0.046	0.221 **	0.053	0.292 **	0.241 **	0.057
FSFI–Lubrication	0.015	0.159 **	0.034	0.244 **	0.195 **	0.048
FSFI–Orgasm	−0.014	0.117 **	0.026	0.239 **	0.201 **	0.037
FSFI–Satisfaction	−0.054	0.167 **	0.029	0.212 **	0.194 **	0.059 †
FSFI–Pain	−0.019	0.133 **	0.045	0.212 **	0.184 **	−0.01
FSFI–Total Score	0.033	0.208 **	0.052	0.305 **	0.256 **	0.067 †
IIEF–Sexual Desire	0.220 **	0.193 **	0.092 †	0.19 **	0.315 **	0.188 **
IIEF–Erectile Function	0.027	0.075	0.04	−0.002	0.227 **	0.067
IIEF–Orgasmic Function	0.036	0.062	−0.002	0.039	0.23 **	0.027
IIEF–Intercourse Satisfaction	−0.016	0.101 †	0.063	0.076	0.248 **	0.058
IIEF–General Satisfaction	−0.066	0.06	0.046	0.032	0.189 **	0.075
IIEF–Total Score	0.023	0.098 †	0.05	0.045	0.291 **	0.079 †

Note: † = *p* < 0.05; ** *p* < 0.001.

**Table 4 healthcare-11-00880-t004:** Person’s correlation matrix between SDEF3 six-factor solution, sociodemographic variables and social desirability (MC-SDS) (*n* = 1773).

	SDEF3 F1	SDEF3 F2	SDEF3 F3	SDEF3 F4	SDEF3 F5	SDEF3 F6
Age	−0.138 **	−0.166 **	0.024	−0.018	0.119 **	−0.12 **
Being in a Relationship	−0.192 **	0.015	−0.027	0.061 †	0.026	−0.058 †
Having Children	−0.127 **	−0.134 **	0.016	−0.053 †	0.071 *	−0.106 **
Education Level	−0.099 **	−0.083 **	−0.05 †	0.028	−0.043	−0.098 **
Political Conservativism (Right winged)	−0.003	−0.051 †	0.014	−0.054 †	0.038	0.058 †
Political Involvement	0.107 **	0.082 *	0.042	−0.01	0.157 **	−0.034
Religious Education	0.03	−0.017	0.025	−0.029	0.033	0.069 *
Religiousness	−0.097 **	−0.151 **	−0.006	−0.099 **	−0.067 *	0.029
Religious Involvement	−0.101 **	−0.159 **	−0.018	−0.109 **	−0.076 *	0.03
Sexual Intercourse in Life	0.076 *	0.125 **	0.020	0.144 **	0.116 **	0.025
Sexual Intercourse in the last six months	0.039	0.174 **	0.039	0.175 **	0.136 **	0.028
Social Desirability (MC-SDS)	−0.212 **	−0.11 **	−0.143 **	−0.057 †	−0.049 †	−0.001

Note: † = *p* < 0.05; * = *p* < 0.01; ** *p* < 0.001.

**Table 5 healthcare-11-00880-t005:** MANCOVAs having Gender and Sexual Orientation as independent variables and SDEF3 six-factor solution as dependent ones (*n* = 1729).

	**Women (*n* = 1088)** **M ± DS**	**Men (*n* = 641)** **M ± DS**		**Δ**	**F_(1,1724)_**	** *p* **	**95% CI**		**Partial Eta^2^**
							**Lower Bound**	**Upper Bound**	
SDEF3–F1. Physical and Contextual	1.09 ± 0.63	1.55 ± 0.72		0.46	70.14	<0.001	−0.984	−0.565	0.039
SDEF3–F2. BDSM	0.84 ± 0.79	0.86 ± 0.81		0.02	-	0.354	−0.306	0.211	-
SDEF3–F3. Taboo	0.07 ± 0.18	0.21 ± 0.36		0.14	20.87	<0.001	−0.165	−0.011	0.012
SDEF3–F4. Bottom	1.49 ± 0.96	1.09 ± 0.9		0.4	-	0.6	−1.242	0.655	-
SDEF3–F5. Top	1.04 ± 0.84	2.38 ± 0.97		1.34	207.89	<0.001	−1.265	−0.674	0.108
SDEF3–F6. Romantic	1.76 ± 0.93	1.79 ± 0.9		0.03	-	0.311	−0.301	0.309	-
	**Heterosexual** **(*n* = 1404)** **M ± DS**	**Bisexual** **(*n* = 152)** **M ± DS**	**Homosexual** **(*n* = 173)** **M ± DS**	**Post Hoc** **Bonferroni**	**F_(1,1724)_**	** *p* **	**95% CI**		**Partial Eta^2^**
							**Lower Bound**	**Upper Bound**	
SDEF3–F1. Physical and Contextual	1.2 ± 0.69	1.5 ± 0.63	1.57 ± 0.74	He < Bi He < Ho	7.64	<0.001	−0.407	−0.161	0.009
SDEF3–F2. BDSM	0.79 ± 0.77	1.25 ± 0.96	1.01 ± 0.79	He < Bi He < Ho Ho < Bi	12.52	<0.001	−0.354	−0.049	0.014
SDEF3–F3. Taboo	0.12 ± 0.28	0.16 ± 0.27	0.16 ± 0.24	-	-	0.382	−0.165	0.011	-
SDEF3–F4. Bottom	1.24 ± 0.92	1.77 ± 0.97	1.82 ± 1.01	He < Bi He < Ho	37.48	<0.001	−1.461	−1.116	0.042
SDEF3–F5. Top	1.48 ± 1.11	1.64 ± 0.96	1.89 ± 1.04	-	-	0.054	−0.072	0.597	-
SDEF3–F6. Romantic	1.78 ± 0.93	1.73 ± 0.88	1.75 ± 0.82	-	-	0.936	−0.126	0.233	-
	**Gender**	**Sexual Orientation**	**M**	**SD**	**F_(1,1724)_**	** *p* **	**Partial Eta^2^**		
SDEF3–F1. Physical and Contextual	Women Men	Heterosexual Bisexual Homosexual Heterosexual Bisexual Homosexual	1.05 1.46 1.03 1.48 1.64 1.78	0.62 0.61 0.6 0.73 0.71 0.68	6.91	0.001	0.008		
SDEF3–F2. BDSM	Women Men	Heterosexual Bisexual Homosexual Heterosexual Bisexual Homosexual	0.77 1.35 1.05 0.82 0.88 1	0.75 0.92 0.78 0.79 1 0.8	3.84	0.022	0.004		
SDEF3–F3. Taboo	Women Men	Heterosexual Bisexual Homosexual Heterosexual Bisexual Homosexual	0.06 0.14 0.1 0.22 0.23 0.18	0.17 0.24 0.24 0.39 0.33 0.24	-	0.118	-		
SDEF3–F4. Bottom	Women Men	Heterosexual Bisexual Homosexual Heterosexual Bisexual Homosexual	1.47 1.81 1.15 0.79 1.63 2.07	0.95 0.97 0.88 0.66 0.96 0.95	54.91	<0.001	0.06		
SDEF3–F5. Top	Women Men	Heterosexual Bisexual Homosexual Heterosexual Bisexual Homosexual	0.98 1.42 1.21 2.44 2.39 2.14	0.82 0.88 0.93 0.97 0.88 0.97	7.21	<0.001	0.008		
SDEF3–F6. Romantic	Women Men	Heterosexual Bisexual Homosexual Heterosexual Bisexual Homosexual	1.77 1.71 1.79 1.8 1.83 1.74	0.95 0.85 0.7 1 0.87 0.9	-	0.700	-		

Note: Age, Relationship Status, and Social Desirability were put as covariates.

**Table 6 healthcare-11-00880-t006:** MANCOVAs having FSFI and IIEF clinical scores as independent variables and SDEF3 six-factor solution as dependent ones (*n* = 1455).

**Women**	**FSFI Functional Score** **(*n* = 647)** **M ± DS**	**FSFI Clinical Score** **(*n* = 289)** **M ± DS**	**Δ**	**F_(1,1083)_**	** *p* **	**95% CI**		**Partial Eta^2^**
						**Lower Bound**	**Upper Bound**	
SDEF3–F1. Physical and Contextual	1.11 ± 0.63	1.07 ± 0.63	0.04	16.76	<0.001	0.085	0.241	0.015
SDEF3–F2. BDSM	0.95 ± 0.82	0.67 ± 0.73	0.28	38.78	<0.001	0.22	0.422	0.035
SDEF3–F3. Taboo	0.08 ± 0.2	0.06 ± 0.16	0.02	-	0.156	−0.007	0.041	0.002
SDEF3–F4. Bottom	1.71 ± 0.94	1.17 ± 0.89	0.54	85.14	<0.001	0.449	0.691	0.073
SDEF3–F5. Top	1.21 ± 0.87	0.8 ± 0.74	0.41	64.47	<0.001	0.335	0.551	0.056
SDEF3–F6. Romantic	1.82 ± 0.92	1.68 ± 0.92	0.14	10.81	0.001	0.08	0.317	0.01
**Men**	**IIEF Functional Score** **(*n* = 430)** **M ± DS**	**IIEF Clinical Score** **(*n* = 89)** **M ± DS**	**Δ**	**F_(1.636)_**	** *p* **	**95% CI**		**Partial Eta^2^**
						**Lower Bound**	**Upper Bound**	
SDEF3–F1. Physical and Contextual	1.56 ± 0.72	1.51 ± 0.73	0.05	12.6	<0.001	0.019	0.019	0.019
SDEF3–F2. BDSM	0.9 ± 0.8	0.69 ± 0.81	0.21	13.07	<0.001	0.147	0.496	0.020
SDEF3–F3. Taboo	0.22 ± 0.35	0.2 ± 0.43	0.02	0.47	0.492	−0.052	0.108	0.001
SDEF3–F4. Bottom	1.11 ± 0.89	0.97 ± 0.95	0.14	4.09	0.044	0.006	0.406	0.006
SDEF3–F5. Top	2.46 ± 0.92	2.01 ± 1.12	0.45	11.49	<0.001	0.154	0.577	0.018
SDEF3–F6. Romantic	1.81 ± 0.91	1.71 ± 0.86	0.1	4.7	0.031	0.021	0.423	0.007

Note: Age, Relationship Status, and Social Desirability were put as covariates.

## Data Availability

Data is unavailable due to privacy or ethical restrictions. Any further request can be directed to the corresponding author.

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
