# Peer review of "Sexual Desire and Erotic Fantasies Questionnaire: Validation of the Erotic Fantasy Inventory Scale (SDEF3) in Italian Adults"

_healthcare, 2023, doi:10.3390/healthcare11060880_

Round 1

Reviewer 1 Report

Reviewing: “Sexual Desire and Erotic Fantasies questionnaire: Validation of 2 the Erotic Fantasy Inventory scale (SDEF3) in Italian adults” (healthcare-2144413). In this study, the authors validate the Sexual Desire and Erotic Fantasies questionnaire (SDEF) part 3 – Inventory of Erotic Fantasies, designed to be an inclusive and sex- positive measure of erotic fantasies’ contents available to all individuals regardless of their gender identities, sexual orientations, relational/romantic status, and sexual behaviors. The manuscript is very interesting and important in this field. In its current form, however, the paper has several major and minor issues that must be dealt with prior to publication. My comments are organized according to the order of the paper:

1. In line 76: " No validation study of the SFQ modified version is available in the literature, although a suggested 6-factor structure was proposed by Dyer et al. (2016) and used by Allen et al. (2022)." Simple search in Google I found- Bartels, R., & Harper, C. A. (2018, August 19). An exploration of the factor structure of Gray et al.’s Sexual Fantasy Questionnaire. https://doi.org/10.31234/osf.io/wxj54  

2. In line 106: “Despite the acknowledgement of the thematical updates, Nese et al. (2021) suggested that the EIQ offers too broad dimensions that are only partially accurate in describing content of fantasies”. The authors want to validate a questionnaire, a more detailed explanation is needed as to why not to use existing questionnaires validated in Italian.

3. Throughout the article, the importance of validating the questionnaire for the clinical aspects is repeated and again. I suggest elaborating more on the clinical significance of the need for this questionnaire.

4. In line 186: To run explorative and confirmative factorial analysis, participants were randomly assigned to two different groups balanced for gender, age, and sexual orientation”. I understand that the authors split into two groups. But I don't understand in the current study what it adds. Usually, two groups are collected differently or different from each other. Please clarify why two groups are needed in the current study!

5. In “Table 3. Person’s Correlation Matrix among SDEF3 domains of the 20-factor solution.” Please arrange the table as it is correct to display Pearson correlations.

6. The authors split the result to three studies. It is not yet understood why the authors split the sample into three studies. I suggest using the word "sample". When researchers use three studies, they should build the method chapter separately and conduct a brief discussion on findings, and after the three studies, write a general discussion.

7. “Figure 1. Diagrams of Gender X Sexual Orientation on SDEF1 factors (MANCOVAs) (n=1729)”- The authors miss one factor!?

8. In the first paragraph of the discussion, I suggest writing the purpose of the study and main findings (please move the 20 factors to the findings chapter). The same is true for the continuation of the discussion part - the 6 factors.

9. Please add clinical implication. 

Author Response

REVIEWER 1

Reviewing: “Sexual Desire and Erotic Fantasies questionnaire: Validation of 2 the Erotic Fantasy Inventory scale (SDEF3) in Italian adults” (healthcare-2144413). In this study, the authors validate the Sexual Desire and Erotic Fantasies questionnaire (SDEF) part 3 – Inventory of Erotic Fantasies, designed to be an inclusive and sex- positive measure of erotic fantasies’ contents available to all individuals regardless of their gender identities, sexual orientations, relational/romantic status, and sexual behaviors. The manuscript is very interesting and important in this field. In its current form, however, the paper has several major and minor issues that must be dealt with prior to publication. My comments are organized according to the order of the paper:

1.In line 76: " No validation study of the SFQ modified version is available in the literature, although a suggested 6-factor structure was proposed by Dyer et al. (2016) and used by Allen et al. (2022)." Simple search in Google I found- Bartels, R., & Harper, C. A. (2018, August 19). An exploration of the factor structure of Gray et al.’s Sexual Fantasy Questionnaire. https://doi.org/10.31234/osf.io/wxj54  

Response: thank you for this comment. Actually, there is a great confusion regarding these measures because many authors called their questionnaire in the same way (SFQ). The manuscript reported by the reviewer (Bartels and Harper, 2018) refers to another measure, again called SFQ, made by Gray et al. In this manuscript is also reported the list of measures called SFQ with a detailed explanation of the differences between each authors (among items and psychometric properties - when explored). We have tried to clarify better in the manuscript. However, the psychometric properties regarding Joyal et al. version remain unexplored (or at least unpublished) as also reported by Bartels and Harper, 2018

  1. In line 106: “Despite the acknowledgement of the thematical updates, Nese et al. (2021) suggested that the EIQ offers too broad dimensions that are only partially accurate in describing content of fantasies”. The authors want to validate a questionnaire, a more detailed explanation is needed as to why not to use existing questionnaires validated in Italian.

Response: Thank for this comment. We have rephrased the paragraph also following the other reviewer suggestions: “Despite the acknowledgement of the thematical updates, Nese et al. (2021) suggested that the EIQ dimensions collect SF based on general characteristics, with the limit that very diverse fantasies may collide in the same statistical dimension. This makes the operationalization of the dimensions on a clinical and research level very difficult. The labels used to explain the themes are not accurate enough to offer a good description of the SF content. Therefore, Nese at al. (2021) claim for the need for a measure that is easier to use and interpret.” Moreover, the reason for a new measure is also emphasized in the following paragraph.

  1. Throughout the article, the importance of validating the questionnaire for the clinical aspects is repeated and again. I suggest elaborating more on the clinical significance of the need for this questionnaire.

Response: thanks to the reviewer. We have tried to improve the discussion on this point.

  1. In line 186: To run explorative and confirmative factorial analysis, participants were randomly assigned to two different groups balanced for gender, age, and sexual orientation”. I understand that the authors split into two groups. But I don't understand in the current study what it adds. Usually, two groups are collected differently or different from each other. Please clarify why two groups are needed in the current study!

Response: Thank you for highlighting this comment. The splitting procedure is commonly reported in literature in order to avoid the possibility of running PCA/EFA and CFA on the same group, which will result in an overfitting of the data (and theoretically inconsistent). Many authors suggest to collect a group and randomly splitting it into 2 consistent group which are comparable for some variables of interest (e.g, main sociodemographic) For an update and a discussion of the literature: https://link.springer.com/article/10.3758/s13428-021-01750-y

  1. In “Table 3. Person’s Correlation Matrix among SDEF3 domains of the 20-factor solution.” Please arrange the table as it is correct to display Pearson correlations.

Response: thank you for this comment. Correlation matrix has been updated.

  1. The authors split the result to three studies. It is not yet understood why the authors split the sample into three studies. I suggest using the word "sample". When researchers use three studies, they should build the method chapter separately and conduct a brief discussion on findings, and after the three studies, write a general discussion.

Response: thank you for this comment. We agree with the reviewer but a previous reviewer (in a previous submission) suggested to report the results in 3 studies. We have deleted the division in 3 studies in order to make it smoother. Regarding using the word “sample” instead of group, we would prefer to use the word “group” because usually “sample” refers to a more rigorous sampling method (not a snowball technique). But if the reviewer prefers us to substitute, we are at your disposal.

  1. “Figure 1. Diagrams of Gender X Sexual Orientation on SDEF1 factors (MANCOVAs) (n=1729)”- The authors miss one factor!?

Response: thank you so much for this comment. Actually figure 1 shows only significant interactions emerged in the MANCOVAs showed in table 5. That is why only 4 graphs appears rather than 6 (as the factors). In order to be clearer, the publisher team should locate the figure 1 after table 5.

  1. In the first paragraph of the discussion, I suggest writing the purpose of the study and main findings (please move the 20 factors to the findings chapter). The same is true for the continuation of the discussion part - the 6 factors.

Response: thank you also for this comment. We have tried to improve the paragraph as suggested.

  1. Please add clinical implication. 

Response: thanks to the reviewer. We have tried to improve the discussion on this point.

Reviewer 2 Report

Overall comments

It is a good study. I believe there are several minor changes the manuscript requires to have a better version of your study. However, there is one comment on the potential cases that may had to have been removed from the sample, which could change (likely not significantly) one trend or two. Yet, if it is the case, it will change most if not all numbers.

I strongly advice to do a cleaner job on the tables. They do not look nice on the proof manuscript. Maybe losing the bold in letters and lines may help. Still, I strongly advice to get your money's worth in the edition work from the Journal team.

Specific comments

Abstract

-          It is unclear when the authors say “differentiate between sexually clinical and functional 20 women and men”

Introduction

-          I believe the entire article may benefit from using SF as an acronym for sexual fantasies considering how many times the term is repeated.

-          Line 52: I suggest separating into a new paragraph after the citation

-          Authors need to make the distinction between sexual fantasies and day dreaming. See

Klinger, E. (2009). Daydreaming and fantasizing: Thought flow and motivation. In K. D. Markman, W. M. P. Klein, & J. A. Suhr (Eds.), Handbook of imagination and mental simulation (pp. 225–239). Psychology Press.

-          Authors use the same acronym to denote the sex and sexual fantasy questionnaires. Moreover, they even include a third questionnaire in the same acronym. This needs to be changed.

-          Line 107: the wording when using “too broad dimensions” is odd and ambiguous.

-          Line 108: THE content of fantasies

-          Lines 130-131: Authors speak about cisgender, and then refer to sexual identities, behaviors and expressions. I believe they mean gender identities, etc. Otherwise, a clarification is required.

-          Authors omit a citation, but their names are given plain as day at the top.

-          The authors do not indicate if the SDEF was also created to have a cross-cultural instrument, as it was mentioned as one of the limitation of previous instruments. If this is not contemplated in the SDEF creation, it should be discussed, probably as a limitation, too.

Methods

-          Authors are encouraged to use MacDonald’s Omega instead of Cronbach’s Alpha to test reliability as it is a better and less biased estimator. See

-          From alpha to omega: A practical solution to the pervasive problem of internal consistency Estimation Thomas J. Dunn*, Thom Baguley and Vivienne Brunsden

-           COMMENTARY ON COEFFICIENT ALPHA: A CAUTIONARY TALE  SAMUEL B. GREEN

-          Authors do not mention anything about individuals who were not sexually active in the last 4 weeks when measuring sexual function with either the IIEF and FSFI. These are criteria to exclude participants as including them inflates the ratings. Only if these were zero would lead to no changes. However, if the authors performed the analyses with them, these cases must be excluded and analyses must be performed again (this is highlighted even by the authors in line 523

-          Authors do not provide a justification as to why they used maximum likelihood, not the interpretation on every CFA indexes, nor do they explain what they do.

Results

-          Table 2 and 4 are perhaps better suited as supplementary.

-          Half of table 3 is unnecessary

-          Authors should do a better job disentangling the content into more paragraphs. Too much info of different nature in the same paragraph.

-          There are several explanations given in this section that should be in the methods.

-          Study 3. Authors should provide a citation to support their rationale when justifying their covariates.

-          If gender diverse individuals are that few, supplementary analyses should still be provided in order to facilitate their inclusion and visibility.

-          There are stats where only one decimal is provided. Two appears to be the standard.

-          I criticized the journal edition team for how incredibly messy the tables look in the manuscript from table 5 on. I strongly advice authors to make their best effort to provide tables that may help them do a better job, though I believe this is the responsibility of the production team. One may think that, after paying them to publish, and that reviewers do this for free, too, that they would do a better job.

-          Graph words do not look pristine when blown up.

Discussion

-          The first paragraph of a discussion requires more gravity to it. Authors could do a better job to entice the reader with more in it.

-          Line 383: I believe the 20 factors would be better displayed through a table

-          Authors need to declare and discuss the rather small gender diverse simple as a limitation of their study.

-          Line 554: I honestly do know if sexual identities, which has been mentioned before, is even a term. I do believe gender identities is.

Note to the authors: I would qualify your work between high and average in terms of Quality of Presentation, Scientific Soundness, and Overall merit. Though, the journal does not give me such score.

Author Response

REVIEWER 2

Comments and Suggestions for Authors

Overall comments

It is a good study. I believe there are several minor changes the manuscript requires to have a better version of your study. However, there is one comment on the potential cases that may had to have been removed from the sample, which could change (likely not significantly) one trend or two. Yet, if it is the case, it will change most if not all numbers.

I strongly advice to do a cleaner job on the tables. They do not look nice on the proof manuscript. Maybe losing the bold in letters and lines may help. Still, I strongly advice to get your money's worth in the edition work from the Journal team.

Specific comments

Abstract

-          It is unclear when the authors say “differentiate between sexually clinical and functional women and men”

Response: Thank you for this comment. We have added “based on FSFI and IIEF cut-off scores)

Introduction

-          I believe the entire article may benefit from using SF as an acronym for sexual fantasies considering how many times the term is repeated.

Response: thank you for this comment. We have used SF were more appropriate according to your suggestion.

-          Line 52: I suggest separating into a new paragraph after the citation

Response: Thank you for this comment, we have modified the manuscript accordingly

-          Authors need to make the distinction between sexual fantasies and day dreaming. See

Klinger, E. (2009). Daydreaming and fantasizing: Thought flow and motivation. In K. D. Markman, W. M. P. Klein, & J. A. Suhr (Eds.), Handbook of imagination and mental simulation (pp. 225–239). Psychology Press.

Response: Thank you for this comment. We have briefly introduced the main differences as follows: “Sexual fantasies (SF) are usually defined as subjective mental imagery and thoughts that are erotic or arousing to the individual while awake (Leitenberg & Henning, 1995). SF are usually finalized to arouse the individual and were distinguished from daydreaming, which is a spontaneous and fanciful series of thoughts not connected to immediate reality and need of satisfaction (Klinger, 2009).”

-          Authors use the same acronym to denote the sex and sexual fantasy questionnaires. Moreover, they even include a third questionnaire in the same acronym. This needs to be changed.

Response: thank you for this comment. We have tried to modify it in order to be clearer for the reader. However many questionnaires have the same name (SFQ) in the literature and this create confusion.

-          Line 107: the wording when using “too broad dimensions” is odd and ambiguous.

Response: Thank for this comment. We have rephrased the paragraph also following the other reviewer suggestions: Despite the acknowledgement of the thematical updates, Nese et al. (2021) suggested that the EIQ dimensions collect SF based on general characteristics, with the limit that very diverse fantasies may collide in the same statistical dimension. This makes the operationalization of the dimensions on a clinical and research level very difficult. The labels used to explain the themes are not accurate enough to offer a good description of the SF content. Therefore, Nese at al. (2021) claim for the need for a measure that is easier to use and interpret.

-          Line 108: THE content of fantasies

Response: Thank you for this comment, we have modified the manuscript accordingly

-          Lines 130-131: Authors speak about cisgender, and then refer to sexual identities, behaviors and expressions. I believe they mean gender identities, etc. Otherwise, a clarification is required.

Response: Thank you for this comment. We used the term sexual identities which includes not only gender identities, but also the one related to sexual orientations and other aspects of sexual expression (e.g., fluidity)

-          Authors omit a citation, but their names are given plain as day at the top.

Thank you so much. We forget to put it back. We have now included it

-          The authors do not indicate if the SDEF was also created to have a cross-cultural instrument, as it was mentioned as one of the limitation of previous instruments. If this is not contemplated in the SDEF creation, it should be discussed, probably as a limitation, too.

Response: Thank you for this comment. Actually it was highlighted at line 140, but we have reinforced it: The current study is inserted in this complex scenario, aiming to provide an updated, extensive, and inclusive measure of SF to be easily adapted to different cultural back-grounds and used  in clinical work in a second phase of the project (cross-cultural validation).

Methods

-          Authors are encouraged to use MacDonald’s Omega instead of Cronbach’s Alpha to test reliability as it is a better and less biased estimator. See

-          From alpha to omega: A practical solution to the pervasive problem of internal consistency Estimation Thomas J. Dunn*, Thom Baguley and Vivienne Brunsden

-           COMMENTARY ON COEFFICIENT ALPHA: A CAUTIONARY TALE  SAMUEL B. GREEN

Response: thank you for this comment. We have provided omega instead of alpha as suggested.

-          Authors do not mention anything about individuals who were not sexually active in the last 4 weeks when measuring sexual function with either the IIEF and FSFI. These are criteria to exclude participants as including them inflates the ratings. Only if these were zero would lead to no changes. However, if the authors performed the analyses with them, these cases must be excluded and analyses must be performed again (this is highlighted even by the authors in line 523

Response: thank you for highlighting this point. This is another confusing mistake we did in the previous reviewing processes in other journal submission. The calculation of total FSFI/IIEF scores and relevant domain scores (on all but the desire domain) were limited to those who have not indicated a zero score (no sexual activity) on any of the FSFI/IIEF items. Scores reported in table are correct, while frequency number were wrong. Sorry for this inconvenient.

-          Authors do not provide a justification as to why they used maximum likelihood, not the interpretation on every CFA indexes, nor do they explain what they do.

Response: thank you so much. If we add here an explanation of the maximum likelihood and all the indexes used it risks being very wide for the manuscript. We have opted for stating that we used the same procedure and indexes for CFA used in another article we have mentioned (that actually has somehow put the basis for this current work).

Results

-          Table 2 and 4 are perhaps better suited as supplementary.

Response: Thank you, we have modified accordingly

-          Half of table 3 is unnecessary

Response: Thank you we have adapted the table

-          Authors should do a better job disentangling the content into more paragraphs. Too much info of different nature in the same paragraph.

Response: thank you for this comment. We have modified the manuscript accordingly

-          There are several explanations given in this section that should be in the methods.

Response: thank you for this comment. We have modified the manuscript accordingly

-          Study 3. Authors should provide a citation to support their rationale when justifying their covariates.

Response: thanks for this comment. We have added a citation as suggested.

-          If gender diverse individuals are that few, supplementary analyses should still be provided in order to facilitate their inclusion and visibility.

Thank you for this comment. Due to the length of the manuscript we opted to avoid supplementary analyses for gender diverse people. A new study Using the SDEF has been dedicated to gender diverse people and sexual orientation other than gay, lesbian and bisexual in a more numerous group.

-          There are stats where only one decimal is provided. Two appears to be the standard.

Response: thank you for this comment. When the last decimal is zero, we have omitted it (as the same when 0 is the only number before decimals. We followed this rule according to the previous journal where we sent the manuscript. But we are available to change it if needed.

-          I criticized the journal edition team for how incredibly messy the tables look in the manuscript from table 5 on. I strongly advice authors to make their best effort to provide tables that may help them do a better job, though I believe this is the responsibility of the production team. One may think that, after paying them to publish, and that reviewers do this for free, too, that they would do a better job.

Response: Thank you so much for this comment. We have tried to improve them as possible to facilitate publication team work

-          Graph words do not look pristine when blown up.

Response: thanks for this comment. We hope that this will be solved in the publishing version.

Discussion

-          The first paragraph of a discussion requires more gravity to it. Authors could do a better job to entice the reader with more in it.

Response: Thank you for this comment. We have tried to improve this part as suggested. We hope that the present version could be clearer for the reader

-          Line 383: I believe the 20 factors would be better displayed through a table

Response: Thanks for this comment. We have followed the suggestion of reviewer 1 to move this part in the results where we think it might be better to remain as list. But we can easily arrange in a table if needed.

-          Authors need to declare and discuss the rather small gender diverse simple as a limitation of their study.

Response: thank you for this comment. This is in line with the abovementioned comment. We have added it into the limitations

-          Line 554: I honestly do know if sexual identities, which has been mentioned before, is even a term. I do believe gender identities is.
Response: Thank you for this comment. We have tried to respond to this in one of the previous comment.

Round 2

Reviewer 1 Report

Comments were made to my satisfaction.

Author Response

thank you